# Vitamin D Status, Calcium Intake and Risk of Developing Type 2 Diabetes: An Unresolved Issue

**DOI:** 10.3390/nu11030642

**Published:** 2019-03-16

**Authors:** Araceli Muñoz-Garach, Beatriz García-Fontana, Manuel Muñoz-Torres

**Affiliations:** 1Department of Endocrinology and Nutrition, Virgen de la Victoria University Hospital, Institute of Biomedical Research in Malaga (IBIMA), 29010 Malaga, Spain; 2Instituto de Salud Carlos III, 28029 Madrid, Spain; 3Instituto de Investigación Biosanitaria (Ibs.GRANADA), 18106 Granada, Spain; bgfontana@fibao.es; 4Centro de Investigación Biomédica en Red sobre Fragilidad y Envejecimiento Saludable (CIBERFES), Instituto de Salud Carlos III, 28029 Madrid, Spain; 5Unidad de Gestión Clínica Endocrinología y Nutrición, Hospital Universitario San Cecilio de Granada, Avenida de la Innovacion, 18016 Granada, Spain; 6Department of Medicine, University of Granada, 18016 Granada, Spain

**Keywords:** calcium intake, dairy products, vitamin D, type 2 diabetes

## Abstract

The relationship between vitamin D status, calcium intake and the risk of developing type 2 diabetes (T2D) is a topic of growing interest. One of the most interesting non-skeletal functions of vitamin D is its potential role in glucose homeostasis. This possible association is related to the secretion of insulin by pancreatic beta cells, insulin resistance in different tissues and its influence on systemic inflammation. However, despite multiple observational studies and several meta-analyses that have shown a positive association between circulating 25-hydroxyvitamin D concentrations and the risk of T2D, no randomized clinical trials supplementing with different doses of vitamin D have confirmed this hypothesis definitively. An important question is the identification of what 25-hydroxyvitamin D levels are necessary to influence glycemic homeostasis and the risk of developing T2D. These values of vitamin D can be significantly higher than vitamin D levels required for bone health, but the currently available data do not allow us to answer this question adequately. Furthermore, a large number of observational studies show that dairy consumption is linked to a lower risk of T2D, but the components responsible for this relationship are not well established. Therefore, the importance of calcium intake in the risk of developing T2D has not yet been established. Although there is a biological plausibility linking the status of vitamin D and calcium intake with the risk of T2D, well-designed randomized clinical trials are necessary to answer this important question.

## 1. Introduction

The incidence of type 2 diabetes (T2D) has increased substantially in recent years related, in part, to higher obesity rates. If current trends continue, more than 642 million people will have diabetes by 2040 [1]. The management and early treatment of T2D are essential to prevent further complications involving loss of quality of life and premature death. It is unclear whether vitamin D deficiency might be contributing to an increased T2D risk [2].

A vast body of evidence associates vitamin D deficiency and T2D [3]. This relationship could be mediated by the direct and indirect effects of vitamin D on glucose homeostasis such as insulin secretion, insulin sensitivity, and systemic inflammation. However, the extent of this relationship and its clinical relevance are not well established. 

There is no doubt that vitamin D homeostasis is of vital importance for skeletal health, being especially important for bone mineralization. Moreover, recent studies have demonstrated that low vitamin D concentrations are related to other pathologic conditions that were not previously considered, such as insulin resistance, T2D, metabolic syndrome and cardiovascular diseases. All these diseases could potentially be developed as a result of vitamin D deficiency [4,5,6].

Moreover, the evidence linking milk and, in particular, calcium intake, insulin secretion and sensitivity has been related to glucose homeostasis in both prediabetes and T2D [7,8,9]. However, the main clinical studies conducted to confirm this hypothesis have yielded inconsistent results. 

The aim of the present review is to summarize the recent evidence linking vitamin D and calcium intake with the development of T2D. We also analyzed different intervention studies with vitamin D supplements to determine their influence on glucose metabolism.

## 2. Methods

We performed a comprehensive literature search on PubMed to identify peer-reviewed articles on vitamin D levels, vitamin D supplementation and T2D prevention published until December 2018. Search strategies included the following search terms: vitamin D intake, vitamin D supplementation, 25-hydroxyvitamin D, calcium intake, dairy products, type 2 diabetes, impaired glucose tolerance, insulin resistance, insulin sensitivity, β-cell function and obesity. 

We included a selection of papers that showed original research articles in humans mainly published in English language; and also in vitro studies, observational studies (prospective and retrospective) and randomized controlled trials. Finally, we reviewed meta-analyses published compiling all studies information. Priority was given to larger studies (according to number of patients included) and the most recent and strongest available evidence.

## 3. Vitamin D and Type 2 Diabetes (T2D)

### 3.1. Vitamin D Physiology and Glucose Homeostasis

The term vitamin D includes vitamin D2 or ergocalciferol and vitamin D3 or cholecalciferol. The main metabolites of vitamin D, which differ in their hydroxylation patterns, are 25-hydroxyvitamin D or calcidiol (25(OH)D) and 1,25-dihydroxyvitamin D3 or calcitriol (1,25-(OH)_2_D_3_). In humans, the main sources of vitamin D come from the skin through the cutaneous synthesis of vitamin D3 and, to a lesser extent, from the intake of foods rich in vitamins D2 and D3 or supplements. Circulating vitamin D is bound to vitamin D binding protein (DBP), which transports it to the liver, there vitamin D25-hydroxylase converts it to 25(OH)D. This form of vitamin D is primarily converted to the most biologically active form, 1,25-(OH)_2_D in the kidneys. This transformation is done by the enzyme 25-hydroxyvitamin D-1alpha-hydroxylase (CYP27B1). The presence of CYP27B1 in multiple tissues, which also express the vitamin D receptor, suggests that vitamin D could play an important function beyond bone metabolism.

Both in vitro and in vivo studies have reported that vitamin D may play an important role in the maintenance of pancreatic beta cell function [10]. This effect could have different explanations. It could be induced by the activation of the vitamin D receptor (VDR) located in pancreatic beta cells. It was suggested by the study results that showed how mice without VDR have impaired insulin secretion [11] and the addition of calcitriol to the culture medium stimulated pancreatic islets and resulted in an increased insulin secretion [12]. 

Moreover, vitamin D could also influence insulin secretion by regulating calcium channel opening and closure. Calcitriol participates as a chemical messenger interacting with different receptors regulating calcium flux in beta cells. They are located on the phospholipid layers of plasma membranes. For this reason, calcium is essential for appropriate insulin secretion by pancreatic beta cells; and, therefore, insufficient vitamin D may alter normal insulin secretion through alterations in calcium flux in beta cells [13,14]. In relation to this, the regulation of the protein calbindin, a calcium-binding protein, by vitamin D may be another mechanism influencing insulin secretion. In addition, preclinical studies show that vitamin D can reduce the hyperactivity of the renin angiotensin system and, thus, improve the functioning of beta cells (Leung PS. Nutrients 2016 [15]).

An adequate vitamin D level can also improve insulin resistance pathways associated with diabetes. It is caused mainly by alterations in calcium flux and concentration through the cell membranes of insulin-responsive tissues [16]. The regulation of extracellular and intracellular calcium concentrations may promote dephosphorylation of glucose transporter-4 (GLUT-4) driving a reduced insulin-stimulated glucose transport [14,17]. The 1,25-(OH)_2_D stimulates the expression of insulin receptors and, therefore, stimulates insulin sensitivity. In addition, calcitriol could also improve insulin sensitivity activating the peroxisome proliferator-activated delta receptor (PPAR-d), a transcription factor that regulates fatty acids metabolism in adipose tissue and the skeletal muscle. Another interesting study indicates that insulin resistance may also be reduced by the specific effects of calcitriol on hepatic lipid synthesis and glucose output, and on skeletal muscle (Leung PS. Nutrients 2016 [15]).

Calcitriol has a central role in a wide variety of metabolic pathways by binding to the VDR, and the measurement of its substrate 25(OH)D is an important marker for health risks. This receptor is expressed in an assortment of cells, such as in the pancreatic beta cells of Langerhans, but also in liver, adipose tissue, and muscle cells [18,19]. The VDR and the 1α-hydroxylase, the enzyme catalyzing calcidiol to calcitriol conversion, are expressed in primary preadipocytes and recently differentiated adipocytes [18]. Therefore, in vitro studies suggest that calcitriol regulates the growth of human adipose tissue and its remodeling. Moreover, fat tissue is a storage site for vitamin D [19]. In contrast, a higher body mass index (BMI) is associated to lower vitamin D concentrations. Vitamin D, a fat-soluble hormone, is sequestered in the adipose tissue and, consequently, only small quantities are available for circulation [20]. On the other hand, since the concentrations of 25(OH)D in serum and adipose tissue are closely related, obesity can reduce serum 25(OH)D through volumetric dilution and the distribution of 25(OH)D in larger fat volumes [21]. 

Vitamin D could also shorten the effects of chronic inflammation, and it is well established that it plays a key role in the pathogenesis of T2D. Therefore, 1,25(OH)_2_D can protect against cytokine-induced apoptosis of beta cells directly regulating the activity and expression of cytokines, with an improvement in insulin sensitivity [22]. Moreover, vitamin D demonstrated the possibility of deactivating inflammatory cytokines associated with insulin resistance and promoting calbindin expression which involves protection from apoptosis [23]. Finally, vitamin D also reduces the accumulation of advanced glycation products in experimental studies [24]. These products are related with the development of T2D complications and have been involved with insulin resistance. Vitamin D functions related with glucose homeostasis are summarized in Figure 1.

### 3.2. Vitamin D Status and Its Relationship with T2D in Cross-Sectional and Longitudinal Studies 

Serum 25(OH)D concentrations have been noticed to be inversely associated with glucose homeostasis, insulin resistance, and beta cell function, and forecast lower risks of both metabolic syndrome and T2D [25,26,27]. Numerous clinical studies have associated vitamin D inadequacy with the development of insulin resistance in different populations, not only in adults [5,28,29] but also in children [30,31].

Consistently, higher baseline 25(OH)D levels have been found to predict better beta cell function and lower glucose levels in subjects at risk for T2D in longitudinal studies [32]. Overall, data from observational studies strongly support an association between low vitamin D status and incidence of T2D [33,34,35].

We discuss in this review the largest prospective articles and some meta-analyses. In 2013, Afzal et al. published the results of a prospective cohort study that included 9841 participants who were followed-up for 29 years. They found an odds ratio for the development of T2D of 1.5 (95% CI 1.33–1.70) between the lowest and the highest quartile of 25(OH)D [34]. More recently, Park et al. measured 25(OH)D levels in a cohort of 903 adults without diabetes or prediabetes, these authors found an inverse dose-response association between 25(OH)D concentration and risk of diabetes. They proposed a target 25(OH)D of 50 ng/mL; higher than the levels previously suggested in other studies, in the attempt to influence and reduce the incidence rate of diabetes [36]. These data are consistent with the levels published recently by Avila-Rubio et al. in postmenopausal women, the authors link values of 25(OH)D > 45 ng/dL in these women with better glycemic indexes measured by homeostasis model assessment (HOMA) [37]. 

The multicenter EPIC-InterAct study measured plasma 25(OH)D metabolites: non-epimeric 25(OH)D_3_, 3-epi-25(OH)D_3_ and 25(OH)D_2_. They identified that plasma non-epimeric 25(OH)D_3_ (the major component of total 25(OH)D) was inversely associated with T2D, whereas 3-epi-25(OH)D_3_ was positively associated with the incidence of T2D, and 25(OH)D_2_ was not associated with T2D [38]. 

Another large cohort was The Melbourne cohort, which included a sample of middle-aged Australians, the authors showed how vitamin D status was inversely associated with the risk of T2D and, apparently, this association cannot be explained by reverse causality [39]. If the association was due to reverse causality, then a much stronger association would be expected to be observed in the first few years of follow-up.

The meta-analysis conducted by Song et al. included 21 observational studies with 76,220 subjects in total; the authors found a 38% lesser risk of developing T2D in the highest baseline reference category of 25(OH)D compared to the lowest one (95% CI 0.54–0.70) [35].

Despite the consistency of these results, all these were observational studies and estimation of causality cannot be completely excluded because of residual confounding agents.

### 3.3. Vitamin D Supplementation and Risk of T2D: Randomized Trials and Meta-Analysis 

In the last decade, more than ten well-designed, randomized trials evaluated the effect of vitamin D3 supplementation on glucose homeostasis in subjects at risk for T2D and showed inconsistent results. We have selected a set of studies that analyzed outcomes related to the objectives of this review. Table 1 summarizes the main results of these studies. 

In Table 1 we describe the main findings of the largest trials. Sollid et al. [40] conducted a randomized clinical trial with approximately 500 prediabetes subjects comparing vitamin D versus placebo for the prevention of T2D. They supplemented with 20,000 IU cholecalciferol weekly and after one year, no significant differences were reported between those receiving vitamin D and those taking placebo in any of the glycemic or inflammatory markers and blood pressure, regardless of baseline serum 25(OH)D concentrations. 

Two years later, Forouhi et al. [41] compared, in another large randomized trial including 340 prediabetes or at risk of developing T2D subjects, the effect of supplementation with cholecalciferol or ergocalciferol (both 100,000 IU/month) versus placebo, for four months. Prediabetes was estimated by the Cambridge Risk Score [42]. Despite vitamin D supplementation, neither ergocalciferol or cholecalciferol, resulted in increased 25(OH)D_2_ and 25(OH)D_3_ concentrations. No differences in HbA1c concentration were found between groups. It is important to point out that only half of the subjects had concentrations of 25(OH)D < 50 nmol/L. This data could influence the results.

Their results are in accordance with previous findings by Davidson et al. [43]. They supplemented a cohort of Latino and African Americans subjects with prediabetes and hypovitaminosis D at baseline for one year. They used a cholecalciferol dose sufficient to raise serum 25(OH)D levels into the upper-normal range versus placebo. They did not find any effect on insulin secretion or sensitivity, nor the proportion of subjects who developed T2D or whose oral glucose tolerance test became normal [43]. 

However, it is possible to find some positive effect on glycemic markers in some studies. In this line, Gagnon et al. [44] reported when they performed a post hoc analysis only including subjects with prediabetes, an improvement in insulin sensitivity indices was observed. Previously, they gave a supplement of calcium carbonate 1200 mg and cholecalciferol 2000–6000 UI daily to subjects with glucose intolerance or recently diagnosed diabetes, but they found no effect on insulin sensitivity or secretion, and beta cell function. So, despite most clinical randomized trials failing to show a favorable effect of vitamin D supplementation on glycemic control, insulin sensitivity indices, and incident T2D [40,41,43,44,45,46,47,48,49] in subjects at risk for diabetes, there is some interesting evidence supporting a beneficial effect of vitamin D on beta cell function. In fact, Mitri et al. [50] reported in 2011 a significant improvement in insulin secretion in 92 prediabetic subjects who were overweight or obese and at risk for T2D. They were supplemented with cholecalciferol 2000 IU daily and calcium carbonate versus placebo for four months. An important restriction of the above described studies is that they were not designed specifically to assess glycemic homeostasis and the results found correspond to post hoc analyses. 

Although not being designed for this purpose, we would like to point out the results of the Vitamin D and Omega-3 Trial (VITAL), a large-scale trial that evaluated high-dose vitamin D supplementation. This study was designed to evaluate the effect of supplementation with vitamin D on incidence of invasive cancer or cardiovascular events versus placebo. Overall, no differences were found between groups. However, in Black Americans a potential beneficial effect was found in cancer mortality [51].

A recent meta-analysis conducted by Rafiq et al. showed an inverse relationship; higher vitamin D concentrations were associated with lower BMI in T2D patients and non-diabetic subjects at risk for T2D. But this association was more pronounced in T2D patients. Moreover, the correlation was directly associated to the BMI quartiles, so the highest BMI quartile had the greatest correlations in both populations, both T2D and non-diabetic [52]. 

Tang et al. [53] published a meta-analysis and did not find an effect of vitamin D supplementation on the incidence of T2D. However, the authors suggested a possible dose-response effect of vitamin D supplementation to improve glucose and insulin metabolism among non-diabetic adults. They postulated a possible benefit of taking vitamin D supplements in higher doses for the primary prevention of T2D.

In summary, studies were very heterogeneous in terms of design, duration, and type of supplement administered and participants characteristics. It is noteworthy that adherence to the treatment would have played a major role in arguing these results.

Recently, the design of a new randomized clinical trial has been published and its results can be a determinant in clarifying many of the uncertainties that exist today. The D2d is a large randomized clinical trial (including participants from 22 sites across the U.S.) hypothesizing that supplementation with vitamin D_3_ daily lowers risk of diabetes in adults with prediabetes [54]. This trial meets people with a large spectrum of diabetes risk, more convenient for testing the underlying hypothesis. D2d trial results are expected to answer two important questions: whether vitamin D supplementation is useful to prevent T2D and how the 2010 expanded American Diabetes Association (ADA) criteria for prediabetes would impact the natural history of this state previous to diabetes.

### 3.4. New Thresholds for the Relationship between Vitamin D and T2D

An important question that has arisen is what 25(OH)D levels are necessary to influence glycemic homeostasis and the risk of developing T2D. Three recent studies have addressed this issue. Von Horst et al. found that optimal 25OHD concentrations for reducing IR were around 50 ng/dL in a randomized controlled study with 81 Asian women [49]. Avila Rubio et al. [37], in a study conducted in women with postmenopausal osteoporosis, suggested that the established goal of reaching a level of 25(OH)D > 30 ng/mL was insufficient to improve glucose metabolism in these population. The data from this study indicates that 25(OH)D > 45 ng/mL are necessary to achieve this goal. These data are consistent with a cohort study of 903 adults of 12 years of duration in non-diabetic population where reaching values of 25(OH)D > 50 ng/mL contributed to reach the maximum benefits to reduce the risk of incident diabetes [36]. Therefore, it is important to establish what 25(OH)D values are necessary to achieve and, even more importantly, maintain all the potential benefits of vitamin D. The currently available studies do not allow us to answer this question with certainty.

## 4. Calcium Intake and T2D 

### 4.1. Mechanistic Studies

To introduce and understand the underlying mechanisms that associates dairy products, and specifically dairy components to T2D prevention, mechanistic studies are essential. Further to the structural role it plays in the skeleton, calcium is an essential electrolyte necessary for many critical biological functions. Calcium may play a key role in a wide range of functions related to glucose homeostasis. Calcium regulates insulin-mediated intracellular processes in specific tissues that respond to insulin, participates in the secretory function of pancreatic beta cells and the phosphorylation of insulin receptors. Calcium also down-regulates specific regulatory genes encoding pro-inflammatory cytokines involved in insulin resistance [16,57,58]. 

Insulin secretion is a calcium-dependent process [59]. Calcium is vital for insulin-mediated intracellular processes in those tissues responding to insulin, such as muscle and fat [16,58]. There is a narrow range of intracellular calcium concentration needed for optimal insulin-mediated functions. When there are changes in intracellular calcium concentrations in insulin-responsive tissues there is a contribution to peripheral insulin resistance [17,60] through a dysregulated insulin signal transduction cascade that leads to a lower glucose transporter activity.

An appropriate range of intracellular calcium concentration is also required for some insulin-mediated activities in tissues such as liver, adipose and skeletal muscle [57]. It is important to maintain relatively low intracellular calcium concentrations in these target tissues to have a beneficial effect on the insulin signal transduction cascade [60] and peripheral insulin sensitivity. In addition, low intracellular calcium attenuates cytokine-induced inflammation, augments vascular relaxation and inhibits platelet aggregation. It is important to keep in mind that calcium intake should be considered in the context of dairy intake and dairy products, which provide other important nutrients besides calcium.

### 4.2. Dairy Intake and T2D Risk

Increased dairy consumption is linked to a lower risk of T2D, but the components responsible for this relation are not well established. The participation of specific dairy products needs to be further studied. Calcium, vitamin D, dairy fat, partially hydrogenated oils and specifically *trans*-palmitoleic acid (a natural *trans* fatty acid found in dairy) are key dairy components. They have been proposed to influence some metabolic pathways implicated in T2D prevention. 

We have previously reported how vitamin D has a direct effect on insulin secretion by binding to VDR in pancreatic beta cells and an indirect effect via the regulation of extracellular calcium [13,57,61]. Moreover, vitamin D effects include suppression of inappropriately prolonged inflammation by modulating secretion of proinflammatory cytokines.

The role of dietary intake of *trans*-palmitoleic acid has been related to an improvement in metabolic regulation, hepatic and peripheral insulin resistance, and suppression of hepatic de novo lipogenesis and lower levels of fasting insulin, C-reactive protein, triglycerides and blood pressure [62,63]. In the Cardiovascular Health Study (CHS), a prospective cohort analysis about dietary intake of *trans*-palmitoleic acid, a significantly and considerably 62% risk reduction of incident T2D has been shown [62]. Moreover, the Multi-Ethnic Study of Atherosclerosis (MESA), a prospective cohort study, showed that dietary intake of *trans*-palmitoleic acid was associated with a 48% lower risk of incident T2D [63]. 

When we analyzed the fat content of dairy, the evidence finds relatively consistent results regarding a beneficial role of fatty dairy products in T2D prevention. But, to date, the differences between low, regular or high fat dairy are less known. Kratz et al. described in a systematic review including observational studies [64], that the majority of studies analyzed inversely associated high-fat dairy products with obesity, T2D and cardiometabolic disease, either significantly or insignificantly. However, the meta-analysis of cohort studies conducted by Alhazmi et al. showed that saturated fat ingestion was not associated with a risk of T2D [65]. 

Furthermore, the evidence regarding the role of specific types of dairy products (milk, yogurt, and cheese) is even more limited. Milk has generally been posted as part of total dairy consumption, and scarce evidence exists on milk particularly. It seems that milk consumption may be associated with a T2D risk reduction, with not well-established differences between regular-fat or whole fat milk and T2D [66,67,68]. The association between cheese consumption and a reduced risk of T2D still needs to be strongly supported because some findings are not statistically significant [68,69,70,71]. Finally, limited evidence suggested a protective role of fermented dairy products in general (including yogurt, cheese, buttermilk, and fermented milk), against T2D [69,71]. 

### 4.3. Observational Studies

More than twenty observational studies regarding calcium intake and T2D prevention were found. Different populations have investigated the association between calcium intake and T2D. The largest are four cohort studies. They were done in the United States (*n* = 83,779 [72], (*n* = 41,186) [73]), and the other two in Asian populations: China (*n* = 64,191) [74], and Japan (*n* = 59,796) [75]. They demonstrated an inverse association between dietary or total calcium intake and T2D risk among women but not in men, in the United States and in China. In the Japanese study, they found an inverse association in subjects with higher vitamin D intake. 

In Korea, a smaller cohort study conducted in rural areas (*n* = 8313) also showed an inverse association between total and vegetable calcium intake and T2D risk among women, as previously reported [76]. Conversely, a relatively short study (*n* = 5200) in Australia did not find an association between dietary calcium and T2D [77]. Additionally, other studies provided mixed results when investigating the association between dairy products and the potential risk of T2D. 

More recently, the Korean Genome study, a prospective cohort community-based trial followed for 10 years, explained the longitudinal associations between dietary calcium intake and the incidence of T2D [9]. The authors also associated dietary calcium intake and serum calcium levels at the baseline survey. They found that higher dietary calcium intake was associated with a lower risk of developing T2D. These results are important for public health and have implications for predicting and preventing T2D development. These findings can provide guidelines for calcium dietary and calcium supplementation. 

At the same time, a Spanish study including more than 500 postmenopausal women without diabetes showed a decrease in fasting plasma glucose and glycated hemoglobin after the intervention. This supplementation consisted of a higher dose of vitamin D3 as part of an enriched milk, providing a daily intake of 600 IU of vitamin D3 and 900 mg of calcium [78]. 

Another population-based study using a prospective survey of 5582 adults, the Australian Diabetes Obesity and Lifestyle Study (AusDiab), showed a significant inverse association between the highest tertial of dairy intake and risk of diabetes in men after a following-up period of five years. They obtained these results after adjustment for confounding variables such as age, sex, energy intake, and other potential confounders (adjusted OR 0.53, 95% CI 0.29–0.96 [79]. This inverse association was non-significant in women (adjusted OR 0.71, 95% CI 0.48–1.05). When the authors analyzed different dairy products (low-fat milk, full-fat milk, yogurt, cheese), they only found significant inverse association with diabetes for low-fat milk (adjusted OR 0.65, 95% CI 0.44–0.94).

The Danish population-based lifestyle intervention study done by Struijk et al. called the Inter 99 Study, explained the association between specific types of dairy products and T2D incidence. They did not find a significant association between total dairy intake and T2D incidence (OR 0.95, 95% CI 0.86–1.06) and when they analyzed specific dairy products and T2D no association was reported. Particularly, cheese and other fermented dairy (including yogurt, and buttermilk) appeared to have a beneficial effect on glucose regulation markers, with an inverse association with fasting plasma glucose and glycated hemoglobin [69].

At the same time, the results of the Whitehall II prospective cohort study of working staff of Civil Service departments, were reported. They followed 4186 subjects for ten year and found that total dairy consumption was not significantly associated with T2D (hazard ratio (HR) 1.30, 95% CI 0.95–1.77). They analyzed all possible dairy (high-fat and low-fat dairy, total milk, yogurt, cheese and fermented products) and were not associated with T2D risk. But, nevertheless, fermented dairy products were significantly associated with an inverse risk of overall mortality [68]. 

In the French population, the Data from the Epidemiological Study on Insulin Resistance Syndrome (DESIR) study analyzed 3435 participants prospectively for nine years. This cohort study showed that global consumption of dairy products, but not cheese, was inversely associated with new impaired fasting glycemia diagnosis and T2D (adjusted OR 0.85, 95% CI 0.76–0.94). When analyzing cheese consumption, they did not find an association with T2D (adjusted OR 0.93, 95% CI 0.82–1.06). Curiously, they reported an inverse relationship between cheese and incident metabolic syndrome (adjusted OR 0.82, 95% CI 0.71–0.95) [70].

Moreover, data from the Nurses’ Health Study II, including 37,038 women followed-up for seven years, evaluated the possible influence of dairy consumption during adolescence with the development of T2D later in adulthood. They adjusted for risk factors present in adolescence. Those women with the highest quintile intake of dairy during adolescence (two servings per day) had a 38% lower risk of T2D. They also adjusted for risk factors appearing in adulthood and still demonstrated a significant inverse association between adolescent dairy intake and T2D (RR 0.73, 95% CI 0.54–0.97). There was a 43% T2D risk reduction in women with high-dairy intakes for a long time (from adolescence to adulthood), highlighting the importance of persistence in dairy consumption. They also found a 25% risk reduction for the highest current dairy consumption (two servings per 1000 kcal), and a 26% and 28% risk reduction with low- and high-fat dairy consumption, respectively [80]. In contrast, the EPIC Study, including 16,835 participants in a nested case-cohort analysis including eight European countries, found no association between total dairy consumption and T2D (HR 1.01, 95% CI 0.83–1.34) [71]. There were an inverse association between the consumption of cheese and fermented dairy products (cheese, yogurt, and thick fermented milk) with T2D (HR 0.88, 95% CI 0.76–1.02 and HR 0.88, 95% CI 0.78–0.99, respectively).

Recently the PURE study was published [81]. It analyzes the association of dairy intake with cardiovascular disease and mortality in 21 countries from five continents. In this study, the dietary intake of dairy products of 136,384 individuals were recorded using country-specific validated food frequency questionnaires. During a follow-up period of 9.1 years, the incidence of cardiovascular events and mortality was evaluated. The authors concluded that higher consumption of dairy products is associated with lower risks of mortality and cardiovascular disease.

To date, there are no well-designed randomized controlled trials (RCTs) that have specifically studied the relationship between dairy products and the risk of incident T2D. However, there is a randomized crossover trial with 12 months follow up that need to be considered. It evaluated the consumption of low-fat dairy (four servings per day) and was associated with better insulin resistance, without negative effects on body weight and lipid profile [82]. 

### 4.4. Systematic Reviews and Meta-Analyses 

Three meta-analyses of prospective cohort studies on dairy products and T2D are worth noting. In a meta-analysis conducted by Tong et al. the highest dairy consumption, compared to the lowest category, significantly reduced the risk of T2D by 14% [67]. They found a significant inverse association for low-fat dairy and yogurt with T2D. This association was not found for high-fat dairy and whole (regular-fat) milk. They described, in a dose-response analysis, how each additional daily serving of total dairy showed a decrease of 6% in T2D risk. Especially per each additional serving of low-fat dairy intake there was a 10% T2D risk reduction. 

Elwood et al. have previously demonstrated in their meta-analysis including four prospective cohort studies on diabetes, that milk or dairy consumption played a protective role against T2D. Per each additional daily serving there was a 4%–9% risk reduction in diabetes incidence [66]. Pittas et al. found in their meta-analysis including mostly similar cohort studies that the highest versus lowest dairy intake (3–5 vs. 1.5 servings per day) was associated with a lower risk of incident T2D [57].

So, we can consider that, to date, there is some strong, consistent, and accumulating evidence about the influence of dairy intake on a reduced the risk of T2D [7].

However, it is important to notice possible confounding factors such as fat content in some dairy products, which can influence the protective effects of calcium [74,83]. Moreover, calcium intake may also depend on other products non-dairy foods (for example tofu, fish, rice, vegetables, and pulses). It is evident that the main source of dietary calcium differs between populations and different cultures.

## 5. Nutritional Recommendations for T2D Prevention

An important body of evidence has shown that dairy products can reduce the risk of T2D significantly and probably in a dose-response way.

The value of having an adequate intake of dairy products should be reinforced especially among those with prediabetes, obesity and metabolic syndrome.

In addition, cultural differences, nutritional habits, economic status and gender are related to the consumption of milk and dairy products. Unfortunately, a large percentage of the adult population, particularly older adults, do not to meet the international recommendations for optimal calcium intake and need to be encouraged to increase daily calcium intake.

Dairy products are largely under-consumed by all age groups and across populations. More than 80% Americans do not meet the minimum dairy requirements of the Dietary Guidelines for Americans (DGA) [84]. The same problems have been identified in other cultures, such as the Chinese population. Their milk intake is still quite low [85].

The amount of calcium needed daily varies by age. The recommendations given by the National Institutes of Health (NIH) proposed a daily intake of 1000 mg for men between 25 and 65 years [86]. This is the same recommendation for women between 25 and 50 years, with an exception for pregnant or lactating women or postmenopausal women not receiving estrogen replacement therapy. They should take 1500 mg/day [86]. For all subjects, men and women over 65 years, the NIH proposes a daily calcium intake of 1500 mg [86]. On the other hand, recently updated the US Institute of Medicine (IOM) recommendation of 1000 mg/day of calcium intake for all adults aged 19–50 years and for men until age 70 years. They recommend 1200 mg/day for women 51 years or older and both men and women aged >70 years [87]. They proposed in their guidelines that calcium-rich foods, especially milk and other dairy products, are the best source of calcium intake because they have showed a higher absorption efficiency. Alternatively, calcium supplementation may help reaching optimal intake for those subjects who cannot take adequate calcium through diet alone.

Although 25(OH)D blood concentration is the most commonly used biomarker to determine vitamin D status, there is no global consensus on what the 25(OH)D thresholds are for vitamin D deficiency or insufficiency. The main guidelines issued by the IOM and the Endocrine Society differ on their classification of vitamin D status. The differences can be explained because of the various populations recognized by the guidelines and the way evidence was described. The IOM guidelines focus on the general healthy population and emphasize on interventional studies. The IOM did not find appropriate evidence linking vitamin D and beneficial effects for non-skeletal outcomes, such as diabetes. Therefore, the IOM argued that a level of 25(OH)D >20 ng/mL is adequate and enough for skeletal outcomes, whereas only low evidence data ratify a higher level. Moreover, the IOM proposed that a level >50 ng/mL should be followed to avoid potential adverse events. In contrast, the Endocrine Society clinical practice guidelines focus on people at high risk for vitamin D deficiency and emphasize more on observational (epidemiological) studies. The latter guidelines determined that 25(OH)D concentrations >30 ng/mL are desirable for optimal skeletal outcomes without suggesting any upper limit to be concern for safety. However, the Endocrine Society guidelines have been criticized for the way they characterized several subgroups as a high-risk population and their wide recommendations for screening for vitamin D deficiency [88]. There is agreement between both guidelines about the requirement to reconsider current recommendations in the future when ongoing randomized trials become available. Thus, two important questions are raised. First, the existence of different thresholds for different beneficial effects. Second, the harmonization of techniques to determine circulating 25(OH)D concentrations to achieve comparable results [89].

## 6. Unsolved Questions

In this review, a large body of evidence has been discussed about the intake of calcium and vitamin D and its association with the incidence of T2D, although the results are inconsistent. To date, several observational studies and randomized trials have been performed including very heterogeneous subject populations. They differ in design and duration, and in which range of vitamin D types and calcium products and various dosing regimens used. Therefore, it seems necessary to clarify what vitamin D levels are needed to obtain a real benefit, if any, on glycemic status, and this concentration is probably higher than recommendations currently focused on obtaining a benefit on bone metabolism. Supplementation with vitamin D at doses around 4000 IU/day may be an option to increase 25(OH)D levels close to 50 ng/mL and improve homeostasis rates of glucose and insulin among non-diabetic subjects. Therefore, no consensus regarding whether the general population needs further supplementation of vitamin D to improve health outcomes has been found.

Furthermore, more research is needed to better understand the role of calcium intake from milk and specific types of dairy products (regular fat, skimmed, fermented, non-fermented) on the incidence of T2D and indices of glucose metabolism.

Establishing if specific populations such as those with prediabetes, the overweight or obese, could obtain significant benefits with nutritional recommendations regarding the intake of calcium and vitamin D has become a matter of special interest.

## 7. Conclusions and Perspectives

We conclude that the current literature is inadequate for drawing firm conclusions about the association between calcium intake and incident T2D, although it appears that a higher consumption of dairy products may be beneficial for glucose metabolism. Moreover, an adequate level of vitamin D may also have a helpful effect on T2D prevention, and a potential dose-response effect is suggested.

Nevertheless, specific studies with a close control of calcium intakes and higher vitamin D supplementation are needed to better understand their effects on glucose and insulin homeostasis.

## Figures and Tables

**Figure 1 nutrients-11-00642-f001:**
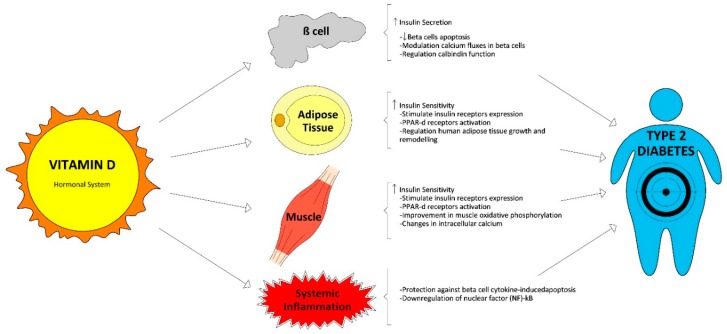
Vitamin D functions related with glucose homeostasis.

**Table 1 nutrients-11-00642-t001:** Clinical trials investigating the association between vitamin D supplementation and risk of type 2 diabetes (T2D).

Study, Year	Country	Population Characteristics (Mean age/age range (years))	Type of Treatment	*N*	Duration (months)	Main Outcomes
LeBlanc et al. D2d Research Group [54] 2018	US	Prediabetes (59)	4000 IU D3/day vs. placebo	2423	36	Not yet published
Forouhi et al. [41] 2016	UK	Prediabetes IFG or IGT or positive diabetes risk score (53)	100,000 IU D2/month or 100,000 IU D3/month vs. placebo	340	4	No effect on HbA1c Improve pulse wave velocity (arterial stiffness)
Wagner et al. [48] 2016	Sweden	Prediabetes or diet-controlled T2D (67.3)	30,000 IU D3/week vs. placebo	44	2	No difference in insulin secretion/sensitivity, beta cell function and glucose tolerance
Tuomainen et al. [47] 2015	Finland	Prediabetes (65.7)	40 μg/day D3 or 80 μg/day D3 vs. placebo	68	5	No difference in glucose homeostasis indicators
Gagnon et al. [44] 2014	Australia	25(OH)D ≤ 22ng/ml at risk of T2D (54)	1200 mg calcium carbonate and 2000–6000 IU D3 day to target vs. placebo	95	6	No difference in insulin secretion/sensitivity and beta cell function. A post-hoc analysis (only prediabetes patients) showed a significant beneficial effect on insulin sensitivity
Sollid et al. [40] 2014	Norway	Prediabetes IFG or IGT (62.1)	20,000 IU D3/week vs. placebo	511	12	No difference in insulin secretion/sensitivity or glucose metabolism, blood pressure or lipid status
Oosterwerff et al. [45] 2014	Netherlands	Overweight, vitamin D deficient subjects with prediabetes (20–65)	Calcium carbonate 500 mg (all) and 1200 IU D3/day vs. placebo	130	4	No difference in insulin sensitivity or in beta cell function. A post hoc analysis (without diabetes patients at baseline), showed a significant increase in the insulinogenic index when 25(OH)D ≥ 60 nmol/L
Salehpour et al. [46] 2013	Iran	Healthy, overweight/obese women (38)	25 μg D3/daily vs. placebo	77	4	No effect in glycemic indices (glucose, insulin, HbA1c and HOMA-IR)
Belenchia et al. [55] 2013	US	Obese adolescents (14.1)	4000 IU D3/day vs. placebo	35	6	Significant effect in fasting insulin, HOMA-IR and leptin-to-adiponectin ratio
Davidson et al. [43] 2013	US	Prediabetes and hypovitaminosis D (52)	D3 to target serum 25OHD level of 65–90 ng/mL vs. placebo	109	12	No difference on insulin secretion/sensitivity or the development of diabetes or returning to normal glucose tolerance
Mitri et al. [50] 2011	US	Prediabetes. At risk for T2D (57)	2000 IU D3/daily vs. calcium carbonate 800 mg/day	92	4	Significant effect in beta cell function and improvement in insulin secretion
von Hurst et al. [49] 2010	New Zealand	Insulin resistance, At risk for T2D 25(OH)D < 20 ng/mL (23–68)	4000 IU D3/day vs. placebo	81	6.5	No difference in FPG, HOMA2%B; C-peptide Significant effect on HOMA IR and insulin
Jorde et al. [56] 2010	Norway	Overweight/Obese; At risk for T2D (21–70)	500 mg calcium/day plus D3, 40,000 IU/week or D3 20,000 IU/week	438	48	No difference in HbA1c, FPG, 2hs PG and HOMA-IR

BMI, body mass index; FPG, fasting plasma glucose; 2hs PG, 2 h plasma glucose; HbA1c, glycated hemoglobin; HOMA-IR, homeostatic model assessment of insulin resistance; IFG, impaired fasting glucose; IGT, impaired glucose tolerance; NAFLD, nonalcoholic fatty liver disease; T2D, type 2 diabetes.

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
