# Peer review of "Vitamin D Status, Calcium Intake and Risk of Developing Type 2 Diabetes: An Unresolved Issue"

_nutrients, 2019, doi:10.3390/nu11030642_

Reviewer 1 Report

General comments

 In this review, the authors summarize the recent evidence linking vitamin D and calcium intake to the development of Type-2 diabetes. The text is well written. However this review is not the first one on the subject and hence lacks some originality. The authors having included a short section on the potential mechanisms that can be invoked, and on dairy products, give the review some novelty. They have also addressed the issue of the calcium intake that is rarely fully covered in observational and CRT studies. The authors should cite the recent review by Lips et al. Vitamin D and type-2 diabetes. (J Ster Biochem Mol Biol 2017;173:280-5) who address the same issue.

 Specific comments

There are a few typographical errors.

Line 22: “despite multiple observational studies and several meta-analyses have shown” should read: “despite that multiple observational studies and several meta-analyses have shown”

Line 24 and page 7 line 1: I suggest changing “which” by “what”.

Author Response

Reviewer #1:

General comments

First, we want to thank you for your comments, which will undoubtedly contribute to improve the quality of our manuscript.

As you recommend, we have included the recent review by Lips et al. Vitamin D and type 2 diabetes. J. Steroid Biochem. Mol. Biol. 2017, 173, 280–285.

Specific comments

There are a few typographical errors.

Line 22: “despite multiple observational studies and several meta-analyses have shown” should read: “despite that multiple observational studies and several meta-analyses have shown”

Line 24 and page 7 line 1: I suggest changing “which” by “what”.

According to your suggestion we have corrected both typographical errors.

Reviewer 2 Report

Review of Nutrients- 463566 [“Vitamin D status, calcium intake and risk of developing type 2 diabetes: an unresolved issue.”]

This MS aims to review the literature on the roles of vitamin D and calcium in the determination of T2DM risk,  especially focussing on what serum 25(OH)D concentration[s] and what intakes of calcium might prove to reduce T2DM risk. The text is reasonably clear and quite easy to follow.

General comments.

1.       The text regularly describes assessment of vitamin D status by measurement of ‘serum vitamin D level[s]. However, it is not measurements of vitamin  D that are referred to but measurements of the 25(OH)D metabolite. Also, laboratories measure the blood content of  such compounds as concentrations, not levels; thus, these terms should be used throughout, reserving the term ‘levels’ for discussion of things like target levels or ‘cut-offs’.

2.       Calcium intakes are reported and discussed as ‘total[s]’.  However, it needs to be made very clear throughout whether such totals are of ’dietary’ calcium alone or include supplemental calcium which many older people may well be taking. This point is raised because the literature suggests that increases in the intake of dietary calcium have beneficial health effects, for example reducing the risks of cardiovascular disease, but that higher intakes of supplemental calcium can have adverse effects on health, especially on cardiovascular health. [e.g. Boucher,2012 - commentary] and so it would be useful to clarify this matter when first discussing what calcium data is acquired and in Discussion, to remind people which of these you are talking about and not just to mention this point in the discussion.

3.       Many specific points mentioned lack clarity, especially on mechanistic matters that are well understood but could be presented as that would help readers new to this field.

Specific queries and comments on the text by section and line numbers. [Note these do not mention those matters covered under the general comments.]

Abstract,  line 30, has not yet been established?; lines 32-3, …are necessary to be able to answer this important question correctly.

Introduction. Line 46, … being especially important for bone mineralization…?; line 55, surely you do not only examine the roles of calcium and vitamin D in dairy products in this study? Please clarify; line 57, you mention looking at findings but only present observational and little mechanistic data,

Methods, line 65, you specify the use of ‘the most relevant papers’ but this is being selective and so you have to say exactly what method you used to judge ‘relevance’ so that section should be more precise and detailed, and  other mentions of the use of ‘relevant’ data. Papers may also need some clarification.

Vitamin D and type 2 diabetes, line 70, the term ‘vitamin D’ does not include all its metabolites, it includes cholecalciferol and ergo-calciferol, the parent compounds of their metabolites, please edit; line 75, specify the specific 25-hydroxylase; line 78-9, why not mention that those tissues activating vitamin to form calcitriol also express the VDR [vitamin D receptor] since that fact  strengthens the argument for expecting calcitriol to have actions in those tissues?  Line 80+, the effect of calcitriol on  beta cells are well known, early reports found vitamin D was necessary for insulin  secretion and that its rapid release through the calcium related mechanisms you mention [lines 85-90] [phase 1] that activate proinsulin  splitting enzymes followed by slower increases in its secretion [phase 2] modulated by gene activation through the nuclear VDR. Many other known effects exist suppressing damaging over activation of various systems [e.g. the renin-angiotensin system [Leung PS’s group], so that some additional detail might help readers; lines 95-100, insulin  resistance may also be reduced by specific effects of calcitriol on hepatic lipid synthesis and glucose output [Leung PS group] and on skeletal muscle; lines 101+, calcitriol is  activated in target tissues  which is an important reason for serum concentrations of its circulating substrate [25(OH)D] so often being a marker of health risks and this should be mentioned; line 107, lower circulating concentrations of 25(OH)D in obesity appear to be due to dilution into expanded fat masses [see refs from Hypponen E group] and intact vitamin D sequestration in fat is another matter altogether; line 113, calcitriol is well known to suppress secretion of  pro- and enhance secretion of anti-inflammatory cytokines [e.g., Hewison M group]; line 118, Figure 1 is a bit simplistic, with nothing on the liver; line 126,  higher baseline 25(OH)D concentrations forecast lesser risks of both MetS and T2DM 10 years later; [e.g. Forouhi NG group]; line 141 ‘….measured plasma contents ‘ of those metabolites rather than analysed them; line 147, ‘… were not explained by reverse causality’ but you should tell readers why not; line 149, I hope that those subjects were not patients as such studies are usually carried out in population based cohorts;  line 150 state whether those categories were at baseline or were the achieved values; line 152 …of causality..; line 155, ….of vitamin D3l…; line 158, what made trials outstanding – how did you decide, and is this the same as ‘most relevant’, if not  then you must state how you chose those reports in the Methods;  line 163,  were there enough deficient subjects to provide adequate power for the detection of the pre-specified outcomes in that subgroup - this point should be clarified; line 181, effect, not action; line 188, a pity not to mention the paper from von Hurst et al, 2010, which was specifically designed for showing reductions in abnormal IR in VitD deficient normoglycemic  subjects and found them if serum 25(OH)D reached 80nmol/l or above;  line 189+ look again at the ViTAL paper rather than the publicity about it as there were reductions in  cancer mortality and other health benefits , especially in Black Americans; line 200, re phrase for clarity; lines 204-5 on non-compliance, you should quote papers where this has been assessed, as it has reached as high as 80% for VitD supplementation; Table 1 is too small  to be legible – try it Landscape instead of Portrait which just might correct this problem.;

Line numbers have vanished after line 213 and further comments are, therefore, by section headings and their line numbers.

4.Calcium intake and T2DM. 4.1, line 2, why only dairy products, fatty spreads and some fish contain VitD3 and in some countries milk, orange juice cereals and non-dairy spreads are fortified, so please clarify what you were [as mentioned in the general comments]. Para 2, line 1, insulin release is a calcium dependent process [and why repeat this in the next para…; 3rd para, last 2 lines, explain why only dairy related intakes need consideration  when many foods,  and most drinking water, contains significant amounts of calcium and  when magnesium intakes area vital for glucose homeostasis and also for vitamin D efficacy since most enzymes, including those in the vitamin D activating cascade, are heavily dependent on Mg for their efficacy, thus, these matters should be mentioned – since Mg deficiency is common you might like to look at magnesium provided by the diet as a factor in T2DM risk; if not now, then in the future;  4.2, para 1,  line 1, increased dairy consumption …?para 2, last sentence, vitamin  Ds effects include suppression of inappropriately prolonged inflammation [see e.g. Hewison M group work]; last sentence, ‘ … by modulating secretion of cytokines influencing inflammation’. page 8, first line, unclear – do you mean ‘…. ..analysis of the effects of [? Dietary or supplemental]  intakes of trans-….;  line 4, again, is it ‘increased intakes’ or ‘supplemental’ ….?; para 2, line 2, what are ‘fat dairy’ products [.. do you mean fatty dairy?]; para 3, line 4 is very unclear, do you mean that these 3 milk products have very different effects from each other, which is how it reads, or not - please clarify; last sentence, a protective role for fermented products in general,  (….);

Section 4.3,  page 8, Observational studies. 2nd Para, last sentence, is that existing T2DM or prospective risk of T2DM – please clarify; para 4, line 3, why say this intervention twice, , try …intervention with a higher [state amount] dose of vitamin …… in an enriched milk supplying a daily ….’; Page 9, Para 1, line 4-5, please specify what ‘fermented dairy’ are for those not in this area; para 2, line 1, study of working…..; line 4, looks as if ‘no’ should be ‘not’; para 4, line 5, still showed rather than also showed?; line 8, …importance of consistently high dairy food consumption? Para 5, line 5, ‘those authors concluded that’  [unless it was you that drew this conclusion].

Section 5.0, page 10, 2nd sentence, ….should be reinforced in those with …..; Para 3, line 3, which  Asians, there are South Asians [Indian continent] but ‘Asians’ usually means Chinese -  please clarify; Para 5, line 1 read oddly, do you mean ‘the amount of calcium needed daily varies …’?; the last sentence mentioning calcium supplementation should include a comment on the current state of the debate on whether supplemental calcium intakes have dose-wise detrimental health effects or not, and do so in the light of whether the present study looked solely at dietary calcium intakes or has data for supplemental calcium intakes – also, if the latter data is available why not analyse and reported on its association with T2DM risks?

Page 11, para 1, last sentence, you might like to add a comment on the evidence accumulating on there being different thresholds for different beneficial effects from better VitD status whether observationally or after supplementation [Scragg R, 2018] and consider mentioning that all such further advice must allow for variability in assay data produced by different methodology for measuring serum 25(OH)D.

Section 7. line 1, maybe inadequate for drawing firm conclusions rather than insufficient, as the numbers of papers are increasing, being over 15000, overall and over a thousand for ‘calcium, diabetes, dietary’ in PubMed.

Author Response

Reviewer #2:

General comments

First, we want to thank you for your comments, which will undoubtedly contribute to improve the quality of our manuscript. In response to your kind suggestions:

1. The text regularly describes assessment of vitamin D status by measurement of ‘serum vitamin D level[s]. However, it is not measurements of vitamin D that are referred to but measurements of the 25(OH)D metabolite. Also, laboratories measure the blood content of  such compounds as concentrations, not levels; thus, these terms should be used throughout, reserving the term ‘levels’ for discussion of things like target levels or ‘cut-offs’.

2. Calcium intakes are reported and discussed as ‘total[s]’.  However, it needs to be made very clear throughout whether such totals are of ’dietary’ calcium alone or include supplemental calcium which many older people may well be taking. This point is raised because the literature suggests that increases in the intake of dietary calcium have beneficial health effects, for example reducing the risks of cardiovascular disease, but that higher intakes of supplemental calcium can have adverse effects on health, especially on cardiovascular health. [e.g. Boucher,2012 - commentary] and so it would be useful to clarify this matter when first discussing what calcium data is acquired and in Discussion, to remind people which of these you are talking about and not just to mention this point in the discussion.

This issue seems of special interest to us. In our review we refer to dietary calcium intake and not to pharmacological supplements. We have modified this point in the revised version of the manuscript.

3. Many specific points mentioned lack clarity, especially on mechanistic matters that are well understood but could be presented as that would help readers new to this field.

As you recommend, we have modified those points that you indicate as specific comments to help new readers.

Specific queries and comments on the text by section and line numbers. [Note these do not mention those matters covered under the general comments.]

Abstract, line 30, has not yet been established?; lines 32-3, …are necessary to be able to answer this important question correctly.

As you recommend, we have made these changes.

Introduction. Line 46, … being especially important for bone mineralization…?;

According to your suggestion, we have modified this sentence.

line 55, surely you do not only examine the roles of calcium and vitamin D in dairy products in this study? Please clarify;

According to your comment we have deleted the statement "(as part of milk and dairy products)"

line 57, you mention looking at findings but only present observational and little mechanistic data,

We have rewritten the paragraph "We also aim to examine data available from different studies that evaluated the effects of vitamin D supplementation in this setting, in an attempt to analyze the available findings for and against the role of vitamin D in the prevention and / or management of alterations of glucose metabolism”. In the revised manuscript the paragraph is modified to “we also analyzed different intervention studies with vitamin D supplements to determine their influence on glucose metabolism”.

Methods, line 65, you specify the use of ‘the most relevant papers’ but this is being selective and so you have to say exactly what method you used to judge ‘relevance’ so that section should be more precise and detailed, and  other mentions of the use of ‘relevant’ data. Papers may also need some clarification.

We have modified the phrase by replacing it for “We included a selection of papers that showed original research in humans mainly.” Priority was given to larger studies (according to number of patients included) and the most recent and strongest available evidence.

Vitamin D and type 2 diabetes, line 70, the term ‘vitamin D’ does not include all its metabolites, it includes cholecalciferol and ergo-calciferol, the parent compounds of their metabolites, please edit

The paragraph has been rewritten according to your suggestions.

The term vitamin D includes to vitamin D2 or ergocalciferol and vitamin D3 or cholecalciferol.  The main metabolites of vitamin D, that differ in their hydroxylation patterns, are 25-hydroxyvitamin D or calcidiol (25(OH)D) and 1,25-dihydroxyvitamin D3 or calcitriol (1,25-(OH)2D3).

Line 75, specify the specific 25-hydroxylase

We have changed it to vitamin 25-hydroxylase.

Line 78-9, why not mention that those tissues activating vitamin to form calcitriol also express the VDR [vitamin D receptor] since that fact strengthens the argument for expecting calcitriol to have actions in those tissues? 

In the revised manuscript we have modified the phrase to “The presence of CYP27B1 in multiple tissues, which also express the vitamin D receptor, suggests that vitamin D may play an important role beyond bone metabolism.”  

Line 80+, the effect of calcitriol on  beta cells are well known, early reports found vitamin D was necessary for insulin  secretion and that its rapid release through the calcium related mechanisms you mention [lines 85-90] [phase 1] that activate proinsulin  splitting enzymes followed by slower increases in its secretion [phase 2] modulated by gene activation through the nuclear VDR. Many other known effects exist suppressing damaging over activation of various systems [e.g. the renin-angiotensin system [Leung PS’s group], so that some additional detail might help readers

Considering your interesting contributions, we have added a new sentence at the end of the paragraph. “In addition, preclinical studies show that vitamin D can reduce the hyperactivity of the renin angiotensin system and, thus, improve the functioning of beta cells”. (Leung PS. The Potential Protective Action of Vitamin D in Hepatic Insulin Resistance and Pancreatic Islet Dysfunction in Type 2 Diabetes Mellitus. Nutrients. 2016 Mar 5;8(3):147.)

Lines 95-100, insulin  resistance may also be reduced by specific effects of calcitriol on hepatic lipid synthesis and glucose output [Leung PS group] and on skeletal muscle.

We have added your comment at the end of the paragraph “Other interesting study indicates that insulin resistance may also be reduced by specific effects of calcitriol on hepatic lipid synthesis and glucose output and on skeletal muscle (Leung PS. 2016).

Lines 101+, calcitriol is  activated in target tissues  which is an important reason for serum concentrations of its circulating substrate [25(OH)D] so often being a marker of health risks and this should be mentioned

We have added your comment “Calcitriol plays a central role in a large variety of metabolic pathways by binding to the VDR and the measurement of its substrate 25(OH)D is an important marker for health risks”.

Line 107, lower circulating concentrations of 25(OH)D in obesity appear to be due to dilution into expanded fat masses [see refs from Hypponen E group] and intact vitamin D sequestration in fat is another matter altogether.

We have completed the paragraph according to your comment. “On the other hand, since the concentrations of 25(OH)D in serum and adipose tissue are closely related, obesity can reduce serum 25 (OH) D through volumetric dilution and the distribution of 25(OH)D in larger fat volumes”. (Hyppönen E, Boucher BJ. Adiposity, vitamin D requirements, and clinical

implications for obesity-related metabolic abnormalities. Nutr Rev. 2018 Sep 1;76(9):678-692.).

Line 113, calcitriol is well known to suppress secretion of  pro- and enhance secretion of anti-inflammatory cytokines [e.g., Hewison M group]

We have included a Hewison M paper in the bibliography (Chun RF, Liu PT, Modlin RL, Adams JS, Hewison M. Impact of vitamin D on immune function: lessons learned from genome-wide analysis. Front Physiol. 2014 Apr 21;5:151)

Line 118, Figure 1 is a bit simplistic, with nothing on the liver

We appreciate your commentary, but we have prepared a simplistic figure to introduce those readers new in this area as an easy to understand summary.

Line 126,  higher baseline 25(OH)D concentrations forecast lesser risks of both MetS and T2DM 10 years later; [e.g. Forouhi NG group]

We have completed the paragraph with your comment “Serum 25(OH)D concentrations has been reported to be inversely associated with glucose status, insulin resistance, beta cell function, and forecast lesser risks of both developing metabolic syndrome and T2DM”.

Line 141 ‘….measured plasma contents ‘of those metabolites rather than analysed them

We have made the suggested change “Measured plasma contents of 25(OH)D metabolites”

Line 147, ‘… were not explained by reverse causality’ but you should tell readers why not

We  have added this commentary to the revised manuscript:  We “If the association was due to reverse causality then a much stronger association would be expected to be observed in the first few years of follow-up”.

Line 149, I hope that those subjects were not patients as such studies are usually carried out in population based cohorts

We have replaced patients by subjects.

Line 150 state whether those categories were at baseline or were the achieved values

We have added the word “baseline” to clarify the sentence: “the authors reported a 38% lower risk of developing T2D in the highest baseline reference category of 25(OH)D compared with the lowest one (95% CI 0.54–0.70).

line 152 …of causality..; line 155, ….of vitamin D3l…;

Both terms have been changed.

Line 158, what made trials outstanding – how did you decide, and is this the same as ‘most relevant’, if not  then you must state how you chose those reports in the Methods.

According to your comments, we have modified the following paragraph: "We have selected the 13 studies that provided the most relevant information for this review. In the revised manuscript we say: “We have selected a set of studies that analyzed outcomes related to the objectives of this review”.  We have also deleted the phrase "We summarize in the present review of the most outstanding trials ....". It has been changed. Now it is written as follows:  "In table 1 we describe the main findings of the largest trials”.

Line 163, were there enough deficient subjects to provide adequate power for the detection of the pre-specified outcomes in that subgroup - this point should be clarified

These comments have been added to the revised manuscript: “Only half of the subjects have concentrations of 25(OH)D<50 nmol/l. This data can influence the results”.

Line 181, effect, not action

We have changed it in the revised manuscript.

Line 188, a pity not to mention the paper from von Hurst et al, 2010, which was specifically designed for showing reductions in abnormal IR in VitD deficient normoglycemic  subjects and found them if serum 25(OH)D reached 80nmol/l or above

Your comment seems particularly interesting to us. We have considered including it in the section "3.4 New thresholds for the relationship between vitamin D and T2D”.

Line 189+ look again at the ViTAL paper rather than the publicity about it as there were reductions in  cancer mortality and other health benefits , especially in Black Americans

According to your comment we have rewritten the paragraph referring to the VITAL study

“This study was designed to evaluate the effect of supplementation with vitamin D on incidence of invasive cancer or cardiovascular events versus placebo. Overall, no differences were found between groups. However, in Black Americans a potential beneficial effect was found in cancer mortality”.  

Line 200, re phrase for clarity;

We have rephased for better comprehension: “However, the authors suggested a possible dose-response effect of vitamin D supplementation to improve glucose and insulin metabolism among non-diabetic adults. They postulated a possible benefit of taking vitamin D supplements in higher doses for the primary prevention of T2D”.

Lines 204-5 on non-compliance, you should quote papers where this has been assessed, as it has reached as high as 80% for VitD supplementation

This remark has been commented on in the revised version of the manuscript.

Table 1 is too small  to be legible – try it Landscape instead of Portrait which just might correct this problem.

We will discuss with the publisher this question so that the table can be readable.

4.Calcium intake and T2DM. 4.1, line 2, why only dairy products, fatty spreads and some fish contain VitD3 and in some countries milk, orange juice cereals and non-dairy spreads are fortified, so please clarify what you were [as mentioned in the general comments].

Your commentaries about vitamin D fortified foods are very interesting, but in our review we refer mainly to calcium contained in dairy products. 

Para 2, line 1, insulin release is a calcium dependent process [and why repeat this in the next para…; 3rd.

According to your commentary we have deleted the first sentence of paragraph 3.

3rd para, last 2 lines, explain why only dairy related intakes need consideration  when many foods,  and most drinking water, contains significant amounts of calcium and  when magnesium intakes area vital for glucose homeostasis and also for vitamin D efficacy since most enzymes, including those in the vitamin D activating cascade, are heavily dependent on Mg for their efficacy, thus, these matters should be mentioned – since Mg deficiency is common you might like to look at magnesium provided by the diet as a factor in T2DM risk; if not now, then in the future; 

Your commentaries on the contribution of calcium from water and other foods as well as the intake of magnesium are truly interesting. We agree that these matters merit a review in the future. However, for this review we focused on calcium intake from dairy products.

4.2, para 1,  line 1, increased dairy consumption

We have corrected that sentence in the revised manuscript.

Para 2, last sentence, vitamin Ds effects include suppression of inappropriately prolonged inflammation [see e.g. Hewison M group work]; last sentence, ‘ … by modulating secretion of cytokines influencing inflammation’

This sentence has been revised: “Moreover vitamin D effects include suppression of inappropriately prolonged inflammation by modulating secretion of proinflammatory cytokines”.

Page 8, first line, unclear – do you mean ‘…. ..analysis of the effects of [? Dietary or supplemental]  intakes of trans-….; 

In this sentence we refer to dietary intake. This sentence has been clarified in the revised version.

Line 4, again, is it ‘increased intakes’ or ‘supplemental’ ….?;

We also refer to ´intakes´; as in the previous sentence.

Para 2, line 2, what are ‘fat dairy’ products [.. do you mean fatty dairy?];

Exactly, we refer to fatty dairy.

Para 3, line 4 is very unclear, do you mean that these 3 milk products have very different effects from each other, which is how it reads, or not - please clarify; last sentence, a protective role for fermented products in general,  (….);

We have modified this sentence to clarify it. We try to explain that not all dairy products have the same effect but there are not well established the differences between products and should be further studied.

Section 4.3,  page 8, Observational studies. 2nd Para, last sentence, is that existing T2DM or prospective risk of T2DM – please clarify

We refer to the potential risk of T2D. We have modified the manuscript to clarify it.

Para 4, line 3, why say this intervention twice, , try …intervention with a higher [state amount] dose of vitamin …… in an enriched milk supplying a daily ….’

According to your comment we have rewritten this paragraph so that it is better understood.

“At the same time, a Spanish study including more than 500 postmenopausal women without diabetes showed a decrease in fasting plasma glucose and glycated hemoglobin after the intervention. This supplementation consisted of a higher dose of vitamin D3 as part of an enriched milk, providing a daily intake of 600 IU of vitamin D3 and 900 mg of calcium”.  

Page 9, Para 1, line 4-5, please specify what ‘fermented dairy’ are for those not in this area

We have specified that fermented dairy were cheese, buttermilk and yogurt.

Para 2, line 1, study of working

We have corrected this sentence in the revised manuscript.

Line 4, looks as if ‘no’ should be ‘not’

We have corrected this sentence in the revised manuscript.

Para 4, line 5, still showed rather than also showed?;

We have corrected this sentence in the revised manuscript.

Line 8, …importance of consistently high dairy food consumption?

We have corrected this sentence in the revised manuscript to clarify that the important issue is persistence in dairy consumption.

Para 5, line 5, ‘those authors concluded that’  [unless it was you that drew this conclusion].

We have corrected this sentence in the revised manuscript.

Section 5.0, page 10, 2nd sentence, ….should be reinforced in those with …..;

We have corrected this sentence in the revised manuscript.

Para 3, line 3, which  Asians, there are South Asians [Indian continent] but ‘Asians’ usually means Chinese -  please clarify

In this statement we refer mainly to the Chinese population.

Para 5, line 1 read oddly, do you mean ‘the amount of calcium needed daily varies …’?;

We have corrected this sentence in the revised manuscript.

The last sentence mentioning calcium supplementation should include a comment on the current state of the debate on whether supplemental calcium intakes have dose-wise detrimental health effects or not, and do so in the light of whether the present study looked solely at dietary calcium intakes or has data for supplemental calcium intakes – also, if the latter data is available why not analyse and reported on its association with T2DM risks?

The debate about whether pharmacological calcium supplements have adverse effects on health outcomes, mainly cardiovascular disease, is a matter of great interest. However, in this review we have considered more appropriate to focus on dietary calcium intake. On the other hand, until we know, whether the pharmacological supplements influence the risk of T2D has not been studied.

Page 11, para 1, last sentence, you might like to add a comment on the evidence accumulating on there being different thresholds for different beneficial effects from better VitD status whether observationally or after supplementation [Scragg R, 2018] and consider mentioning that all such further advice must allow for variability in assay data produced by different methodology for measuring serum 25(OH)D.

According to your suggestion we have included a sentence at the end of the paragraph. “Thus, two important questions are raised. First, the existence of different thresholds for different beneficial effects. Second, the harmonization of techniques to determine circulating 25OHD concentrations to achieve comparable results” (Scragg R. Emerging Evidence of Thresholds for Beneficial Effects from Vitamin D Supplementation. Nutrients. 2018 May 3;10(5).

Section 7. line 1, maybe inadequate for drawing firm conclusions rather than insufficient, as the numbers of papers are increasing, being over 15000, overall and over a thousand for ‘calcium, diabetes, dietary’ in PubMed.

According to your comment we have modified this statement.